# Viscosity Plane-Wave UltraSound (Vi PLUS) in the Evaluation of Thyroid Gland in Healthy Volunteers—A Preliminary Study

**DOI:** 10.3390/diagnostics12102474

**Published:** 2022-10-13

**Authors:** Diana-Raluca Petea-Balea, Carolina Solomon, Delia Doris Muntean, Ioana-Teofana Dulgheriu, Cristina Alina Silaghi, Sorin Marian Dudea

**Affiliations:** 1Department of Radiology, “Iuliu Hațieganu” University of Medicine and Pharmacy, 400012 Cluj-Napoca, Romania; 2Department of Endocrinology, “Iuliu Hațieganu” University of Medicine and Pharmacy Cluj-Napoca, 8 Victor Babes Street, 400012 Cluj-Napoca, Romania

**Keywords:** viscosity, shear wave elastography, thyroid gland, healthy subjects

## Abstract

Viscosity and elasticity represent biomechanical properties of soft tissues that suffer changes during the pathophysiological alterations of the tissue in various conditions. This study aimed to determine average viscosity values for the thyroid gland and to evaluate the potential influences of age, gender and body mass index (BMI), using a recent technique Viscosity Plane-wave UltraSound (Vi PLUS). A total of 85 healthy Caucasian volunteers (56 women and 29 men, median age of 29 years, range 17–81 years) were included in this prospective monocentric study conducted between January 2022 and March 2022. Thyroid viscosity was measured using the SuperSonic MACH 30^®^ Ultrasound system (Aixplorer, SuperSonic Imagine, Aix-en-Provence, France), equipped with a curvilinear C6-IX transducer that allows simultaneous quantification of the viscosity and stiffness. The mean thyroid viscosity measurement value was 2.63 ± 0.47 Pa.s. No statistically significant differences were detected between the left and the right lobes of the thyroid gland. A significant positive correlation was found between thyroid viscosity and elasticity (r = 0.685, *p* < 0.0001). There was no statistically significant correlation between body mass index (BMI) and thyroid gland viscosity and elasticity values (r = 0.215, *p* = 0.053; r = 0.106, *p* = 0.333). No correlation between viscosity and gender was established (*p* > 0.05). Vi PLUS represents a new and promising ultrasonographic technique that can provide helpful information for evaluating the thyroid parenchyma, similar to elastography. The effect of the potential confounding factors on thyroid viscosity was negligible, except for BMI.

## 1. Introduction

Various ultrasound methods for evaluating the mechanical proprieties of biological tissues have been widely studied and integrated into daily medical practice throughout the last decades [1]. Among these methods, shear wave elastography (SWE) has proved to be a reliable, non-invasive imaging method that allows for the quantitative evaluation of tissue stiffness [2,3].

Similar to other ultrasonic techniques based on acoustic radiation impulses, SWE considered that biological tissues are purely elastic, neglecting viscosity [1]. Nevertheless, biological tissues demonstrate both elastic and viscous proprieties [1,4]. Previous studies, focused mostly on liver pathologies, have shown that ignoring tissue viscosity leads to errors in elasticity measurements and may bias information crucial for diagnostics [1,5,6]. Since dispersion links to the frequency dependence of the speed of shear waves (SW) and the attenuation of SWs, evaluation of the dispersion proprieties of SWs can be used as an indirect method for measuring viscosity [5,7,8].

The foundation of elastography imaging modalities is that pathological conditions like inflammation and tumors alter tissue elasticity and viscosity. These changes are reflected in alterations in the mechanical viscoelastic properties of soft tissues [5].

Viscosity Plane-wave UltraSound (Vi PLUS) 2D imaging performed via the new Aixplorer MACH 30 system (SuperSonic Imagine, Aix-en-Provence, France) represents a 2D imaging mode that provides visualization and quantification of tissue viscosity. The result is a local measurement of the tissue viscosity at each point of interest in an organ, expressed in pascal second (Pa.s) [6,9].

Thyroid stiffness has been extensively studied using elastography methods, including SWE and strain elastography, and has become a commonly used parameter in evaluating thyroid parenchyma in focal and diffuse pathologies [2,3,10,11]. So far, no general reference values for thyroid viscosity have been established. Therefore, the goal of this study was to determine normal reference viscosity and elasticity values for the thyroid gland and to evaluate the influences of age, gender, and body mass index (BMI). A second objective was to establish a correlation between the viscosity values and the values obtained using 2D Shear-Wave Elastography PLUS (2D-SWE.PLUS).

## 2. Materials and Methods

### 2.1. Study Population

A total of 85 healthy Caucasian volunteers (56 women and 29 men, median age of 29 years, range 17–81 years) were included in this prospective monocentric study conducted between January 2022 and March 2022 at the Radiology Department, Emergency County Hospital, Cluj-Napoca, Romania. The subjects were selected from patients referred for neck ultrasound for non-thyroid pathologies (sialadenitis, lymphadenopathy, temporomandibular joint disorders, etc.). All the healthy volunteers had normal clinical findings, and normal ultrasound B mode, and color Doppler findings regarding the thyroid gland. Patients with a history of thyroid disorders (autoimmune, inflammatory, or thyroid nodules), thyroid hemiagenesis, or previous exposure to radioiodine therapy were excluded.

This study was approved by the Local Ethics Committee and complied with the World Medical Association Declaration of Helsinki, revised in 2000, Edinburgh. Written consent was obtained from all the subjects before study entry.

### 2.2. Viscosity and Shear-Wave Ultrasound Technique

Viscosity and stiffness measurements of the thyroid parenchyma were performed via the new SuperSonic MACH^®^ 30 Ultrasound system (Aixplorer, SuperSonic Imagine, Aix-en-Provence, France). This study involved a single researcher with five years of experience in sonography.

Vi PLUS mode analyzes the shear wave propagation speed at various frequencies to provide information about shear wave dispersion within tissues. Acquisitions of viscosity measurements were done concurrently with the stiffness measurements and utilizing the same methodology since the Vi PLUS mode is coupled with the SWE mode and cannot be applied independently.

The subjects were examined in a supine position with the neck extended, facilitated by the position of a small pillow under the shoulders. No patient preparation was needed. An initial assessment of the thyroid parenchyma was performed using a high-frequency linear transducer (SuperLinearTM SL10-2) to evaluate thyroid volume and exclude unknown pathologies.

Thyroid viscosity and stiffness measurements were performed using the curvilinear transducer (C6-1X) in the sagittal plane for each thyroid lobe to avoid motion artifacts produced by the carotid arteries and trachea. First, we obtained an ultrasound B-mode image with the best gain setting. The probe was placed perpendicularly on the skin surface, and an abundant amount of gel was used to avoid precompression. The measurements were taken while the patient was holding his breath, avoiding deep inspiration or the Valsalva maneuver, which can impact the stiffness of the thyroid parenchyma.

The Vi PLUS mode was activated, and a region of interest was placed at about 1.5–2 cm beneath the skin, with the Vi PLUS box positioned over an area of homogeneous parenchyma, free of large vessels or other structures that might interfere with the measurements. A 5 mm diameter Q-Box placed in the center of the Vi PLUS box was used to obtain the measurements after the image had stabilized for 3 s. The quality control standard for image acquisition was represented by the stability index (SI). If the SI was greater than 90% and the standard deviation was less than 10% of the determined mean viscosity value, the measurement was deemed valid (Figure 1). The acquisition was performed for each thyroid lobe three times to obtain three valid viscosity measurements. The three measures were averaged, resulting in a single value used in subsequent analyses.

### 2.3. Statistical Analysis

The statistical analysis was performed with a dedicated software: MedCalc Version 20 (MedCalc Software 127 Corp., Brunswick, ME, USA). Descriptive statistics were computed—for normally distributed quantitative variables (means and standard deviation) and for non-normal distributed variables (median values and range). Categorical variables were presented as percentages and numbers. One-way analysis of variance (ANOVA) was used to assess the difference between the means of several subgroups. For all the tests, the statistical significance was defined as *p* < 0.05.

## 3. Results

Demographic statistics for the 85 healthy volunteers included in our study are summarized in Table 1.

No significant differences were found between viscosity and stiffness values of the right and left lobes of the thyroid gland in healthy volunteers (*p* = 0.524; *p* = 0.910) (Table 2). Consequently, the mean measurements of the right and left thyroid lobes were included in the analysis for each healthy volunteer.

The mean viscosity value of normal thyroid parenchyma was 2.63 ± 0.47 Pa.s, and the mean stiffness value was 15.89 ± 4.25 kPa (Table 3).

There was no statistically significant difference between mean values of viscosity and stiffness regarding gender groups (*p* = 0.501; *p* = 0.655, respectively), (Table 4). No statistically significant difference was found regarding viscosity between age groups (*p* = 0.958). In the age group between 30–50 years, the SWE values were slightly higher (17.22 ± 5.96) in comparison with the age groups between 17–29 years (15.56 ± 3.84), respectively 51–81 years (15.72 ± 3.73), but did not reach statistical significance as the *p* value was 0.413 (Table 5).

There was no statistically significant correlation between BMI and thyroid gland viscosity and stiffness values (r = 0.215, *p* = 0.053; r = 0.106, *p* = 0.333).

The thyroid gland viscosity and stiffness values were significantly lower in the normal weighted volunteers in comparison with overweighted volunteers (2.52 ± 0.44 Pa.s versus 2.79 ± 0.47 Pa.s, *p* = 0.009; 15.12 ± 3.90 kPa versus 17.11 ± 4.55 kPa, *p* = 0.035) (Figure 2, Table 6).

A significant positive correlation was found between viscosity values and stiffness values for thyroid parenchyma (r = 0.685; *p* < 0.0001) (Figure 3).

## 4. Discussion

Ultrasound SWE has proved to be a helpful diagnostic tool in differentiating between benign and malignant thyroid nodules, especially when combined with Thyroid Imaging Reporting Data System (TIRADS) and has a significant role in identifying the thyroid nodules that can be followed with imaging techniques, reducing the number of fine needle aspiration (FNA) [3,11,12,13]. Additionally, previous studies have shown that SWE has the potential to detect changes in patients with diffuse thyroid diseases, such as autoimmune thyroiditis, acute thyroiditis and Riedel thyroiditis [2,10,14]. A study by Fukuhara et al. showed that SWE measurements were significantly higher in patients with chronic autoimmune thyroiditis in comparison with healthy volunteers [15]. However, SWE techniques were focused mostly on evaluating the differences between benign and malignant thyroid nodules, omitting the viscous component of the parenchyma [12].

Vi PLUS is a recent ultrasound technique that allows the quantification of information regarding shear wave dispersion within a tissue [5]. The principles of wave propagation, where the dispersion is characterized as a compound expression of the poroelastic and microstructural media proprieties controlled by the complex fibrous multiscale microstructure of the stroma, give rise to one of the main characteristics of viscoelastic tissues [16]. The viscosity of the environment seems to play an important role, affecting wave phase velocity, which is dependent on frequency, and wave amplitude which is conditioned by geometric factors. These changes are described in a highly viscous environment. Neglecting viscosity leads to bias for the estimation of stiffness, since the effect of wave dispersion is ignored [16].

In a study by Sugimoto et al. [5], alterations in tissue viscosity correlated to the degree of inflammation in patients with liver fibrosis. Thus, dispersion slope, which reflects viscosity, may provide additional pathophysiological insight into diffuse or focal changes in thyroid parenchyma, especially in the differential diagnosis between focal areas of acute inflammation and focal areas of chronic inflammation. Rianna et al. [17] evaluated thyroid viscoelastic proprieties only on experimental models, showing that malignant thyroid cells tend to be constant regarding viscosity and elasticity proprieties, regardless of substrate stiffness. In comparison, normal thyroid cells become more rigid as the substrate stiffness rises, and the dynamic viscosity exhibits a similar pattern.

In light of the existing studies on the liver parenchyma, it seems that the best use of viscosity lies in determining tissue inflammation and not tumors [5,6,18,19]. Therefore, the usefulness of viscosity in the evaluation of the thyroid gland as well might be directed more towards thyroid acute inflammation and not detect or characterize tumors.

To date, only a few studies have evaluated the potential of this recent ultrasound technique. Muntean et al. [9,20] prospectively evaluated the capability of Vi PLUS in establishing reference values for the parotid and submandibular glands in healthy subjects and also evaluated the functional changes in major salivary glands in healthy subjects. Popa et al. [5] studied the role of viscosity using Vi PLUS mode in patients with liver steatosis, fibrosis, and inflammation in patients with non-alcoholic fatty liver disease (NAFLD) and concluded there is an independent association between viscosity values and liver stiffness values.

To the best of our knowledge, no research has been done to ascertain healthy individuals’ thyroid gland viscosity measurements so far. Establishing the reference viscosity value of thyroid parenchyma is the first step in verifying this novel approach and it also may facilitate early detection of thyroid abnormalities, especially in diffuse inflammatory pathologies.

In our study, we performed three measurements for each thyroid lobe, similar to the procedure followed by Muntean et al. [9].

No considerable differences were found regarding viscosity and stiffness between the left and the right thyroid lobes, therefore the mean viscosity and stiffness values for thyroid parenchyma in healthy volunteers were considered to be 2.63 ± 0.47 Pa.s, respectively 15.89 ± 4.25 kPa. The mean values for thyroid stiffness were higher in our study compared to the values reported in previous studies [21,22]. Arda et al. [21] found in a prospective study that stiffness values of thyroid range between 1–24 kPa, with a mean stiffness of 10.97 ± 3.1 kPa. One study evaluated thyroid stiffness in the pediatric population and showed that the SWE values are lower (10.9 ± 1.78) in comparison with the adult population [13].

Variability regarding the elasticity of the thyroid parenchyma has been previously reported. It may be related to various technical factors such as vendor-specific implementations of SWE mode and increased compression of the thyroid parenchyma, also known as preload [23]. Another possible explanation could be the fact that we used the curvilinear transducer instead of the linear transducer for measuring the elasticity values, at the time of the study the Vi PLUS mode and SWE mode were available only on this type of transducer.

In our study, viscosity and stiffness were not significantly affected by gender. This is in accordance with previous studies that failed to confirm a correlation between gender and thyroid gland stiffness [21,24].

The age group between 30–50 years included in our study presented a mean thyroid elasticity value slightly higher compared with the younger and older age groups but did not reach statistical significance. This incidental finding might be related to the small number of subjects included in this age group.

The thyroid gland viscosity and stiffness values were higher for the patients with an abnormal body mass index. Herman et al. [24] assessed the stiffness of the neck anatomical structures in a normal population using SWE and found that BMI and weight had no significant influence on thyroid stiffness.

A significant positive correlation was found between viscosity values and SWE values for thyroid parenchyma.

Although the diagnostic capabilities of viscosity in identifying acute inflammatory changes have been proved in previous studies on liver, breast and prostate [16], from the clinical point of view the utility of this new parameter is still questionable due to the risk of overdiagnosis.

Combining viscosity with elasticity could result in a more robust imaging approach that provides a complementary understanding of thyroid alterations, especially in diffuse inflammatory conditions.

There were several limitations in this study. First, we did not assess intra- and interobserver reproducibility, all measurements being performed by a single observer. However, recent studies published by our group [20,25] have shown good intra- and interobserver reproducibility of applying Vi PLUS method in calculating viscosity to other soft tissues with similar depth to thyroid parenchyma. Second, no blood tests were performed to confirm normal thyroid function for the healthy volunteers. Third, this study had a relatively small number of subjects. However, we tried to include patients from all age groups, but there was an imbalance in gender distribution with a predominance of female subjects. Fourth, we used the curvilinear transducer to perform all the measurements, given the fact that the Vi PLUS mode was only available on this type of transducer. Although higher frequency transducers are required to determine parenchymal structural changes, this preliminary study’s main goal was to collect quantitative data regarding thyroid viscosity. We used the stability index (SI) greater than 90% to make sure that the image acquisition process was correctly done.

To determine the diagnostic role of this novel technique, further studies with larger series of subjects should be performed to compare the viscosity values of normal and pathologic thyroid tissue.

## 5. Conclusions

In this preliminary study, we used Vi PLUS mode and SWE mode embedded in the new ultrasound machine to determine the mean viscosity and stiffness values of thyroid parenchyma in healthy volunteers. Vi PLUS could represent an important tool that can provide relevant information in respect of the viscous proprieties of biological tissues and open the gates for a new revolutionary field in ultrasound imaging.

## Figures and Tables

**Figure 1 diagnostics-12-02474-f001:**
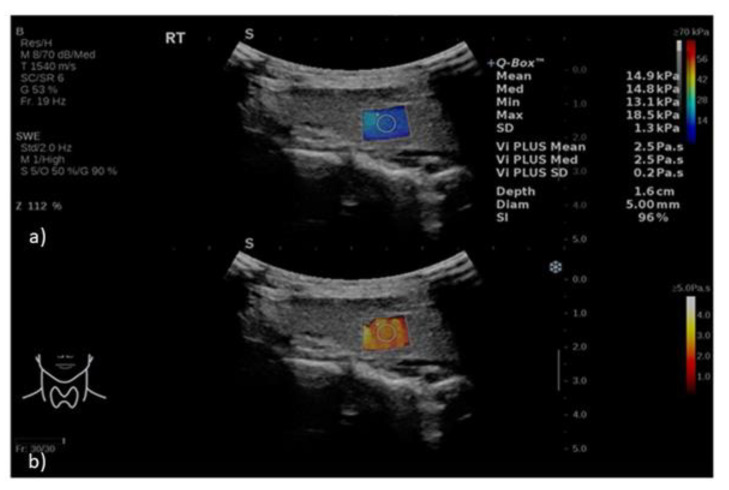
2D-SWE.PLUS (**a**) and Vi PLUS (**b**) measurements obtained in the right thyroid lobe of a healthy volunteer. Vi PLUS box represents a duplicate of the SWE box and for each parameter, a color scale map is displayed. For the SWE mode (range 0 to 70 kPa), low elasticity is coded in blue and high elasticity is coded in red. On Vi PLUS mode (range 0 to 5 Pa.s), low viscosity is color-coded in red, while high viscosity is depicted in white-yellow colors. Simultaneously, quantitative assessment is displayed on the right side of the image for Vi PLUS (expressed in Pa.s) and for SWE (expressed in kPa).

**Figure 2 diagnostics-12-02474-f002:**
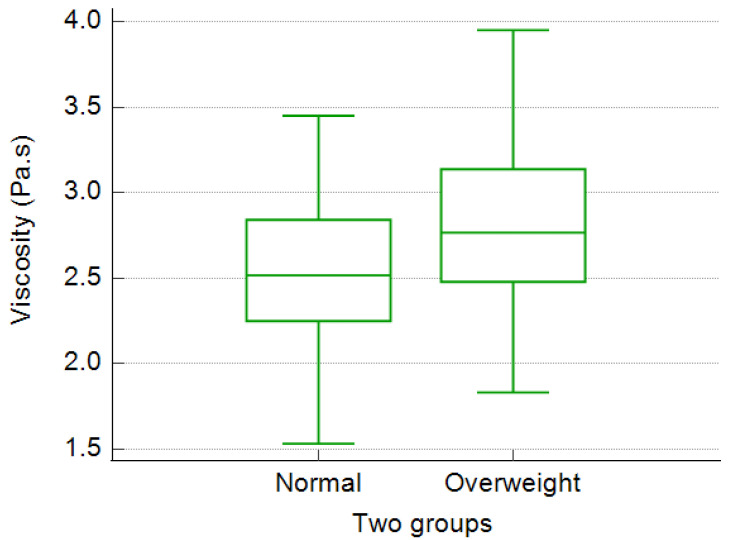
Boxplot diagrams illustrating the differences in mean normal viscosity values of thyroid parenchyma between normal weighted and overweighted groups.

**Figure 3 diagnostics-12-02474-f003:**
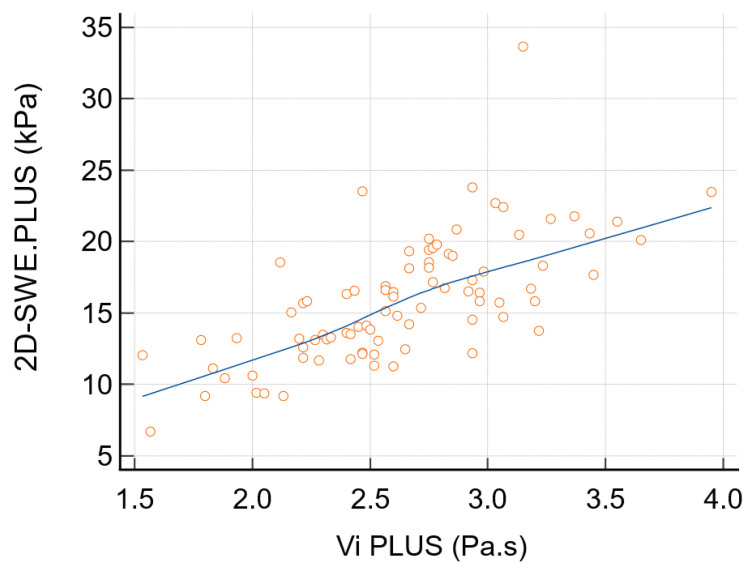
Correlation between the mean viscosity values and stiffness values of thyroid parenchyma in healthy volunteers (blue line—trend line).

**Table 1 diagnostics-12-02474-t001:** Demographic characteristics of the study group.

Descriptor	N (%)/Median (Range)
Total no subjects	85
Female	56 (65.9%)
Male	29 (34.1%)
Age (years)	29 (17–81)
BMI groups	
Normal weight range (18.5–24.9)	51
Overweight range (> 25)	33

N = number of subjects; BMI = body mass index.

**Table 2 diagnostics-12-02474-t002:** Mean viscosity and stiffness values of the right and left thyroid lobes.

		Right Lobe	Left Lobe	
**Viscosity (Pa.s)**	Mean	2.73	2.64	***p* = 0.525**
	SD	1.12	0.54	
	95% CI	2.48–2.97	2.52–2.76	
**SWE (kPa)**	Mean	**15.88**	**15.80**	***p* = 0.911**
	SD	4.24	5.40	
	95% CI	14.95–16.80	14.61–16.98	

SD = Standard deviation, CI = Confidence interval.

**Table 3 diagnostics-12-02474-t003:** Mean viscosity and stiffness values of thyroid parenchyma in healthy volunteers.

	Viscosity (Pa.s)	SWE (kPa)
**Mean** **± SD**	2.63 ± 0.47	15.89 ± 4.25
**95% CI**	2.52–2.73	14.98–16.81
**Min**	1.53	6.7
**Max**	3.95	33.65

SD = Standard deviation, CI = Confidence interval, Min = Minimum, Max = Maximum.

**Table 4 diagnostics-12-02474-t004:** Mean viscosity and stiffness values of the thyroid parenchyma in the healthy volunteers group according to gender.

		Male	Female	
**Viscosity (Pa.s)**	Mean	**2.67**	**2.60**	***p* = 0.501**
	SD	0.59	0.39	
	95% CI	2.50–2.85	2.48–2.73	
**SWE (kPa)**	Mean	**15.60**	**16.04**	***p* = 0.655**
	SD	4.23	4.29	
	95% CI	14.03–17.18	14.91–17.18	

SD = standard deviation.

**Table 5 diagnostics-12-02474-t005:** Mean viscosity and stiffness values of the thyroid parenchyma in healthy volunteers group based on the age distribution.

		Age Group (17–29)	Age Group (30–50)	Age Group (51–81)	
**Viscosity** **(Pa.s)**	**Mean**	**2.63**	**2.65**	**2.60**	** *p* ** **= 0.958**
	**SD**	0.48	0.50	0.43	
	**95% CI**	2.49–2.76	2.41–2.90	2.40–2.81	
**SWE (kPa)**	**Mean**	**15.56**	**17.22**	**15.72**	** *p* ** **= 0.413**
	**SD**	3.84	5.96	3.73	
	**95% CI**	14.35–16.77	15.03–19.41	13.87–17.57	

SD = Standard deviation, CI = Confidence interval.

**Table 6 diagnostics-12-02474-t006:** Mean viscosity and stiffness values of the thyroid parenchyma in the healthy volunteers group according to BMI.

		Normal Weighted (BMI = 18.5–24.9)	Overweighted (BMI > 25)	
**Viscosity (Pa.s)**	**Mean**	**2.52**	**2.79**	***p* = 0.009**
	**SD**	0.44	0.47	
	**95% CI**	2.40–2.65	2.63–2.95	
**SWE (kPa)**	**Mean**	**15.12**	**17.11**	***p* = 0.035**
	**SD**	3.90	4.55	
	**95% CI**	13.97–16.27	15.67–18.55	

SD = Standard deviation, CI = Confidence interval, Min = Minimum, Max = Maximum.

## Data Availability

The data is available only by request.

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
