# Peer review of "Viscosity Plane-Wave UltraSound (Vi PLUS) in the Evaluation of Thyroid Gland in Healthy Volunteers—A Preliminary Study"

_diagnostics, 2022, doi:10.3390/diagnostics12102474_

Round 1
Reviewer 1 Report
The paper describes a simple yet interesting study as it concerns a novel technique in thyroid elastography, namely the assessment of viscosity.
General remarks: I would welcome more details on the viscosity as a feature evaluated in clinical practice. I understand it is novel and the data is scarce, but better description of physical background would be appreciated.
I understand the enthusiasm of the authors for the new technique and elastography itself, but the discussion should be more balanced. For example, the authors stated (lines 206-7): “Also, previous studies have shown that SWE has the potential to detect subtle changes in patients with diffuse thyroid diseases, such as autoimmune thyroiditis [2,14]”. From the point of view of an endocrinologist, the classic B-mode ultrasound is even too sensitive in the detection of changes to thyroid parenchyma when chronic thyroiditis begins. In fact we face the problem of overdiagnosing Hashimoto’s disease, and there is the discussion ongoing in some countries as the treatment should not be started that early. So the changes in patients with autoimmune thyroiditis are not that subtle and the last thing we need is to detect them earlier.
The limitations of the study are well identified and described.
The language should be revised (e.g.: don’t you think that ‘acoustic’ and ‘radiation’ do not go well together – line 40? They denote quite different physical phenomena) . There is also a number of typing errors.
Detailed remarks:
Table 4 clearly shows there was no significant difference in SWE between 3 studied age groups (p=0.413). I presume it was assessed with ANOVA test. Yet the authors claim “In the age group between 30-50 years, the SWE values were significantly higher (17.22±5.96) in comparison with the age groups between 17-29 years (15.56±3.84), respectively 51-81 years (15.72±3.73), p<0.05” – lines 150-151.
Reviewer 2 Report
1. As there was a positive correlation between elasticity and viscosity, elasticity information should be enough to characterize thyroid malignancy and additional viscosity information does not add new information. Clarify?
2. The authors need to perform the same measurements using the elasticity-only mode to show the difference in elasticity with or without consideration of viscosity.
3. The authors need to perform some reproducibility studies.
4. The subjects seem younger than the typical thyroid patient population. How does age affect viscoelasticity? The authors may consider dividing the subjects group into two: younger versus older.
5. Figure 1 caption, SWE mode: Don’t mention fibrosis instead of mention low and high elasticity;
6. Table 1: Add race
7. Table 2: Add left and right viscoelasticity values
8. Line 178: Were both elasticity and viscosity significantly different?
Author Response
Response to Reviewer 2 Comments
Dear Reviewer,
Thank you for your time and effort spend reviewing our manuscript. As a result of the extensive review of the manuscript, we started by abbreviating the title – „ Viscosity Plane-wave UltraSound (Vi PLUS) in the evaluation of thyroid gland in healthy volunteers – A preliminary study.” and also replaced in the manuscript the word „elasticity” with „stiffness”, which seems to be more appropriate. In the following, we will try to give an eloquent answer to your observations.
_____________________________________________________________________________
Point 1: As there was a positive correlation between elasticity and viscosity, elasticity information should be enough to characterize thyroid malignancy and additional viscosity information does not add new information. Clarify?
Response 1: Thank you for pointing out this aspect. In this study, it was not our purpose to diagnose malignancies but to assess the viscosity of the thyroid gland in healthy volunteers. We included in the Discussion section, lines 248-251, the following paragraph: “In light of the existing studies on the liver parenchyma, it seems that the best use of viscosity lies in determining tissue
inflammation and not tumors [5,6,18,19]. Therefore, the usefulness of viscosity in the evaluation of the thyroid gland as well might be directed more towards thyroid acute inflammation and not detect or characterize tumors.“
Point 2: The authors need to perform the same measurements using the elasticity-only mode to show the difference in elasticity with or without consideration of viscosity.
Response 2: Thank you for the very good observation. At the time of study inclusion, on the available ultrasound machine – Super Sonic MACH® 30 Ultrasound system (Aixplorer,SuperSonic Imagine, Aix-en-Provance, France), the viscosity could only be measured with the convex transducer, simultaneously with the elasticity measurement, as a derivate. We have triedto express this better by introducing the following additions in the Material and Methods section, line 90: “Acquisitions of viscosity measurements were done concurrently with the elasticity measurements and utilizing the same methodology since the Vi PLUS mode is coupled with the SWE mode and cannot be applied independently.”
Point 3: The authors need to perform some reproducibility studies.
Response 3: Thank you for the pertinent and relevant observation. Unfortunately, due to the pilot nature of our study, we were unable to assess intra- and interobserver reproducibility regarding viscosity measurements. We acknowledge this limitation at the end of the Discussion section, lines 303-307 and we have tried to express this better by introducing the following additions: “First, we did not assess intra - and interobserver reproducibility, all measurements being performed by a single observer. However, recent studies published by our group [20,25] have shown good intra and interobserver reproducibility of applying the Vi PLUS method of calculating viscosity to other
soft tissues with similar depth to thyroid parenchyma.” Definitely, intra- and interobserver reproducibility of viscosity values for thyroid parenchyma is a topic for future studies.
Point 4: The subjects seem younger than the typical thyroid patient population. How does age affect viscoelasticity? The authors may consider dividing the subjects group into two: younger
versus older.
Response 4: Thank you for the observation. As shown in the Material and Methods section, the-patients were divided into 3 age groups: young adults (17-29 years), middle age adults (30-50 years), and seniors (51-81 years). We did not study children. There were no statistically significant differences between mean values of viscoelasticity regarding gender groups. This information is highlighted in the Results section, lines 156-162.
Point 5: Figure 1 caption, SWE mode: Don’t mention fibrosis instead of mention low and high elasticity;
Response 5: Thank you for the good observation. We have done the changes and replaced fibrosis, respectively absence of fibrosis with low elasticity and high elasticity in Figure 1, lines 119-120–“For the SWE mode, low elasticity is coded in blue and high elasticity is coded in red.”
Point 6: Table 1: Add race
Response 6: Thank you for the suggestion. This aspect is mentioned now in the Materials and Method section – Study population – line 67 : “A total of 85 healthy Caucasians volunteers (56 women and 29 men, median age of 29 years, range 17-81 years) were included in this prospective monocentric study conducted between January 2022 and March 2022 at the Radiology Department, Emergency County Hospital, Cluj-Napoca, Romania.”
Point 7: Table 2: Add left and right viscoelasticity values
Response 7: Thank you for the pertinent observation. A table with the mean values of viscosity and elasticity corresponding to the right and left thyroid lobe has been added in the Results section.
Table 2. Mean viscosity and SWE values of the right and left thyroid lobes in healthy volunteers
| Right lobe | Left lobe | |||
| Viscosity (Pa.s) | Mean SD |
2.73 1.12 |
2.64 0.54 |
p=0.525 |
| 95% CI | 2.48-2.97 | 2.52 -2.76 | ||
| SWE (kPa) | ||||
| Mean SD |
15.88 4.24 |
15.80 5.40 |
p=0.911 | |
| 95% CI | 14.95-16.80 | 14.61-16.98 |
Point 8: Line 178: Were both elasticity and viscosity significantly different?
Response 8: Thank you for the very good observation. The values of both elasticity and viscosity
were significantly different. The elasticity values are now mentioned in the Results section, line
194 – “The thyroid gland viscosity and elasticity values were significantly lower in the normal
weighted volunteers in comparison with overweighted volunteers (2.52±0.44 Pa.s versus 2.79
±0.47 Pa.s, p=0.009; 15.12±3.90 kPa versus 17.11±4.55 kPa, p=0.035).
______________________________________________________________________________
We wish to warmly thank you for your expert, thoughtful, and very pertinent observations, which
made us realize the omissions we made and, we estimate, greatly helped us improve the paper.
With gratitude,
The Authors

Round 2
Reviewer 2 Report
The author should use elasticity instead of stiffness. Stiffness is very general term. Viscosity is also represented as stiffness. Elasticity is more appropriate. In the tables and figures, replace SWE with elasticity. The author is using SW to estimate elasticity and viscosity.
show a fitted line in the fig. 3
Author Response
Reply and comment for reviewer:
We are extremely grateful to the reviewer for the thorough review of our work.
- As per the observation and suggestion:
“ The author should use elasticity instead of stiffness. Stiffness is a very general term. Viscosity is also represented as stiffness. Elasticity is more appropriate. In the tables and figures, replace SWE with elasticity. The author is using SW to estimate elasticity and viscosity”
We definitely agree that viscosity is included in the stiffness of a material and that SW is used to estimate both elasticity and viscosity. We would, however, like to submit the following reply:
For more than a decade, shear wave US has been used to assess the consistency of the liver, in chronic liver disease. The method (Fibroscan, transient unidimensional elastography) produces results in kilopascal and has become, nowadays, the standard for the assessment of liver fibrosis severity. All the papers on the topic (more than 3000 on Pubmed) describe the physical property of the liver assessed by this method as “liver stiffness”.
To better explain our point of view, we would also like to point out the following paragraph:
“Hooke’s law of elasticity is an approximation that states that the extension of a material is directly proportional to the applied stress, σ = EÉ›, where σ is the stress applied to the material, and É› is the strain induced in the material. Stiffness (E) is expressed in kilopascals (kPa) and represents the resistance of material to deformation. While stiff materials, such as concrete, exhibit low strain even at high stress, soft materials such as biological soft tissues exhibit large strain even at low stress.”
(Mueller S, Sandrin L - Liver stiffness: a novel parameter for the diagnosis of liver disease – Hepat Med. 2010; 2: 49–67)
As kPa is a unit of pressure, it is directly proportional (and actually directly measures) a body’s stiffness.
The elasticity modulus (Young’s modulus) is directly related to the stiffness. Elasticity, understood as the ability of a deformed material body to return to its original shape and size when the forces causing the deformation are removed, is inversely proportional to stiffness. It is also influenced by viscosity.
Therefore, when relating to results expressed in kilopascals, we chose to use stiffness, as being the tissue property directly measured by the units, and a term widely used as such in the literature.
Should the reviewer consider the change as imperatively necessary, we are ready to replace “stiffness” with “elasticity modulus”.
Please advise.
- As per the suggestion:
“In the tables and figures, replace SWE with elasticity”.
We are extremely grateful for noticing this inconsistency!
Indeed, in the table headers, we should refer to tissue properties.
Therefore, taking into account the comments above, we feel it would be appropriate to replace (where needed) SWE with either “stiffness” or “elasticity modulus”.
Please advise.
- A best fit line has been inserted in figure 3, as suggested.